# Sinonasal Stent Coated with Sustained-Release Varnish of Mometasone Furoate Inhibits Pro-Inflammatory Cytokine Release from Macrophages: An In Vitro Study

**DOI:** 10.3390/pharmaceutics15031015

**Published:** 2023-03-22

**Authors:** Alessandra Cataldo Russomando, Doron Steinberg, Irith Gati, Ronit Vogt Sionov, Ron Eliashar, Michael Friedman, Menachem Gross

**Affiliations:** 1Department of Otolaryngology-Head and Neck Surgery, Hadassah Medical Center, Jerusalem 9112102, Israel; 2The Biofilm Research Laboratory, The Institute of Biomedical and Oral Research (IBOR), The Faculty of Dental Medicine, Hebrew University of Jerusalem, Jerusalem 9112102, Israel; 3Institute for Drug Research, School of Pharmacy, The Hebrew University of Jerusalem, Jerusalem 9112102, Israel; 4The Faculty of Medicine, The Hebrew University of Jerusalem, Jerusalem 9112102, Israel

**Keywords:** chronic rhinosinusitis, cytokine, nasal stent, steroids, local drug delivery, sustained-release varnish

## Abstract

The aim of the study was to develop a sustained-release varnish (SRV) containing mometasone furoate (MMF) for sinonasal stents (SNS) to reduce mucosa inflammation in the sinonasal cavity. The SNS’ segments coated with SRV-MMF or an SRV-placebo were incubated daily in a fresh DMEM at 37 °C for 20 days. The immunosuppressive activity of the collected DMEM supernatants was tested on the ability of mouse RAW 264.7 macrophages to secrete the cytokines’ tumor necrosis factor α (TNFα) and interleukin (IL)-10 and IL-6 in response to lipopolysaccharide (LPS). The cytokine levels were determined by respective Enzyme-Linked Immunosorbent Assays (ELISAs). We found that the daily amount of MMF released from the coated SNS was sufficient to significantly inhibit LPS-induced IL-6 and IL-10 secretion from the macrophages up to days 14 and 17, respectively. SRV-MMF had, however, only a mild inhibitory effect on LPS-induced TNFα secretion as compared to the SRV-placebo-coated SNS. In conclusion, the coating of SNS with SRV-MMF provides a sustained delivery of MMF for at least 2 weeks, maintaining a level sufficient for inhibiting pro-inflammatory cytokine release. This technological platform is, therefore, expected to provide anti-inflammatory benefits during the postoperative healing period and may play a significant role in the future treatment of chronic rhinosinusitis.

## 1. Introduction

Chronic rhinosinusitis (CRS) is among the most common chronic diseases worldwide and has been reported in all age groups around the world [1]. CRS is broadly defined as a chronic inflammatory disease of the nasal and paranasal sinus mucosa where symptoms have persisted for more than 12 weeks [2]. The pathophysiology of CRS stems from a group of related disorders, making it multifactorial, but it is mostly considered to be an inflammatory disease of the upper airways [3]. CRS has been reported to significantly decrease the quality of life of people suffering from it [1]. The less severe symptoms include nasal congestion, post-nasal drip, and facial pressure, while the more severe quality-of-life-impeding consequences are bodily pain, dysgeusia/dyssomnia, social dysfunction, and even ischemic heart disease and chronic heart failure [1,2]. The estimated incidence of CRS is up to 16% in the United States population, with an estimated total economic burden in healthcare costs that exceeds $30 billion per year [4].

The current treatment options for patients with CRS are antibiotics, steroids (topical or oral), and surgery. While medical treatments do offer relief for most patients, for the ones they do not, endoscopic sinus surgery is commonly performed. Unfortunately, postoperative inflammation, polyposis, and adhesions of the nasal mucous lining often compromise the surgical outcome, which then requires further interventions. Patients who fall into this medical framework are referred to as “failure cases”, and a different treatment option for this group is steroid-eluting sinus stents. These stents are inserted into the nasal cavity during the surgery, and they gradually release a steroid that helps to prevent stenosis (narrowing) of the sinus openings and provide anti-inflammatory benefits during the postoperative healing period [5]. These steroid-eluting stents have been found to be significantly effective, leading to fewer postoperative complications, lower rates of scarring and recurrent polyp formation, sinus inflammation, and higher rates of sinus patency [6].

Currently, there are two FDA-approved steroid-eluting sinus implants available: PROPEL and SINUVA. PROPEL is composed of a bioabsorbable polylactide-co-glycolide polymer coated with corticosteroid mometasone furoate. It works by slowly releasing the drug into the surrounding mucosa in a controlled manner [7]. SINUVA is also made up of bioabsorbable polymers containing mometasone furoate [8]. The most common complications reported for the PROPEL family of stents include postoperative infections (such as sinusitis), followed by displacement of the stent. No adverse events were reported for the SINUVA stent [9].

Macrophages are indispensable key players in both innate and adaptive immunity and are generally considered one of the first lines of defense against pathogens [10]. In nasal polyps (NPs), the M2 macrophage phenotype is elevated and demonstrates impaired phagocytic activity against *Staphylococcus aureus*, which might be an underlying cause for the staphylococcal colonization frequently found in patients suffering from CRS with nasal polyps (CRSwNP) [11].

Neutrophils, another key component of the immune system, were found to play a pathogenic role in CRSwNP [12], and importantly, tissue neutrophilia was associated with a poor response to corticosteroid therapy in patients with CRSwNP [13].

Eosinophils are considered the hallmark of nasal polyps. Nasal polyps show increased levels of mediators important for eosinophil accumulation and survival (e.g., IL-5 and IL-13) in comparison to healthy sinonasal tissue [14]. Patients with eosinophilic nasal polyps often had more severe sinus inflammation (determined by sinus CT scans and nasal endoscopy) as compared to those with non-eosinophilic nasal polyps [15]. Moreover, tissue eosinophilia can be used as a predictor of nasal polyp recurrence following surgery and might serve as an indicator of more recalcitrant disease [16].

TNFα, a pleiotropic cytokine possessing pro-inflammatory properties, is produced by several cell types (epithelial cells, T-lymphocytes, and macrophages) and induces the release of the cytokines IL-6, IL-10, and IFNγ [17]. TNFα recruits and accumulates eosinophils via the upregulation in the nasal polyps of the chemokine eotaxin and the adhesion molecules VCAM-1 and VCAM-2 on the fibroblasts [18]. TNFα plays a primary role in the maintenance of inflammatory reactions, [18] and its serum level is elevated in CRSwNP [19]. TNFα mRNAs and protein levels were found to be higher in the nasal polyps than in the inferior turbinate tissues [20].

IL-6, a pleiotropic cytokine and known inflammation regulator, induces B-cell proliferation and activation, as well as neutrophil recruitment [21]. In CRSwNP, IL-6 mRNAs and protein levels are overexpressed in the nasal polyps when compared to normal mucosa [22]. IL-6 increases sinonasal epithelial cell proliferation after epithelial damage and affects ciliary functions by increasing ciliary beating [23]. This activity is important in the restoration of the barrier’s integrity after epithelial wounding.

The anti-inflammatory cytokine, IL-10, protects the host from excessive tissue damage during its defense against pathogens [24]. IL-10 mRNAs and protein are upregulated in CRSwNP in comparison to normal mucosa and CRS patients without nasal polyposis [25]. Several pathogens have developed mechanisms to upregulate IL-10 during the invasion, resulting in a pro-infectious, immunosuppressed environment. For instance, the immune complexes formed upon the binding of the staphylococcal protein, A (SpA), to immunoglobulins induces IL-10 production [26]. This, in turn, results in the suppression of staphylococcal enterotoxin B-induced IL-5, IL-13, IFN-γ, and IL-17 production in nasal polyp cells [26]. When IL-10 is overexpressed in CRSwNP, the production of pro-inflammatory cytokines is suppressed, resulting in the prevention of pathogen elimination and the maintenance of chronic inflammation [25].

In several countries, mometasone-releasing stents such as PROPEL and SINUVA are not available because of their high cost; therefore, cheaper and more available technology is needed. Herein, we propose using a sustained-release varnish (SRV) containing the steroid mometasone furoate (MMF) on sinonasal stents (SNS), which, in the future, might be used to reduce the mucosal inflammation in the sinonasal cavity for a long period of time. SRV technology has many pharmacologic advantages, including the ability to release drugs slowly from matrixes. There are also benefits, such as prolonged localized exposure to the drug at the site of infection, which reduce systemic side effects and produce better penetration of the drug [27]. Furthermore, repeated medication application is not necessary for SRV technology, which importantly eliminates the issue of patient compliance.

MMF was selected as the drug incorporated into the SRV since it is commonly used in nasal sprays for the treatment of CRS, and it is used in the two above-described steroid-eluting stents. To evaluate the long-term anti-inflammatory effect of SRV-MMF, an in vitro study containing inflammatory cells and cytokines related to CRS disease was conducted.

## 2. Materials and Methods

### 2.1. Sustained-Release Varnish (SRV) Preparation

The SRV-MMF varnish was formulated according to methods previously described by Friedman et al. [27], with the pharmaceutical adaptations of dissolving 31.79 mg of mometasone furoate (MMF) (M2354, Tokyo Chemical Industry, Tokyo, Japan), 2.1749 g of Eudragit RL (Hercules Inc., Wilmington, DE, USA), 0.3209 g of polyethylene glycol 400 (PEG400; 202398; Sigma, St. Louis, MO, USA), and 0.649 g of hydroxypropylcellulose (Klucel EF, Ashland Specialty Ingredients, Switzerland) in 5 mL of absolute ethanol, resulting in the formation of a dry film containing 1.0% mometasone furoate, 10.1% PEG400, 68.5% Eudragit RL, and 20.4% Klucel EF. The placebo varnish (SRV-Placebo) was prepared identically to the experimental SRV, omitting the active ingredient from the formulation.

### 2.2. Coating of the Stents

For this study, the sinus stent used was one (S.T.S Medical Ltd., Misgav, Israel) that is Food and Drug Administration (FDA)-approved and is a removable un-medicated sinus stent.

The sinus–nasal stents (SNS), ArchSinus (S.T.S Medical. PCT/IL2014/050466 filed: 23 May 2014; US Provisional 62/084,831 filed 26 November 2014), were cut into 1 cm segments. Each SNS segment was immersed in 70% ethanol for 1 h, washed three times in sterile double distilled water (DDW), dried, and then coated by dipping them with the varnish MMF and varnish placebo and allowed to dry at room air. The coating was repeated three times for each stent segment. The amount of SRV-MMF was calculated by weighing the SNS segments before and after coating with the SRV-MMF, resulting in 40–50 mg films per segment equivalent to 0.4–0.5 mg of MMF.

### 2.3. Measuring the Biological Activity of MMF Using Different Cellular In Vitro Systems

Mometasone furoate (Tokyo Chemical Industry, Tokyo, Japan) was prepared in DMSO at 10 mg/mL and analyzed for immunosuppressive activity in three different cellular systems at 0.1, 1, and 10 μg/mL. The controls were equivalent to DMSO concentrations without mometasone.

#### 2.3.1. RAW 264.7 Cell Cultures

The mouse RAW 264.7 macrophage cell line (ATCC^®^ TIB-7™) was grown in Dulbecco’s modified Eagle medium (DMEM; D5796, Sigma, St. Louis, MO, USA) supplemented with 10% heat-inactivated fetal calf serum (FCS; F9665, Sigma, St. Louis, MO, USA), 2 mM L-glutamine (03-020-1B, Biological Industries, Beit HaEmek, Israel), 100 U/mL penicillin, and 100 μg/mL streptomycin (03-031-5C, Biological Industries, Beit HaEmek, Israel) and maintained at 37 °C in a humidified atmosphere of 95% air/5% CO_2_ (Binder CO_2_ incubator, TEquipment, Long Branch, NJ, USA). By means of a cell scraper (08-100-241, Fisherbrand, FisherScientific, Shanghai, China), the macrophages were detached from the tissue culture plates (430167, Corning, Incorporated, Kennebunk, ME, USA). The RAW 264.7 cells were then seeded at a density of 400,000 cells per well in 200 µL of DMEM with 10% FCS in 96-well flat-bottomed tissue culture-grade plates (3596, Corning Incorporated, Kennebunk, ME, USA). The following day, the medium was replaced with 100 µL of a fresh DMEM without FCS in the absence or presence of various concentrations of prednisone (P6254, Sigma, St. Louis, MO, USA) or mometasone furoate (M2343, Tokyo Chemical Industry, Tokyo, Japan). After a 30 min incubation at 37 °C, lipopolysaccharide (LPS; L4516, Sigma, St. Louis, MO, USA) in 100 µL of DMEM with 10% FCS was added to a final concentration of 10 ng/mL. Control macrophages received 100 µL of DMEM with 10% FCS without LPS. Following a 24 h incubation period at 37 °C in an atmosphere of 95% air/5% CO_2_, the plates were centrifuged at 1500 rpm for 10 min, and the supernatants were collected to determine the cytokine content (TNFα, IL-6, and IL-10) using the ELISA technique described below (Section 2.5). It is well-known that the LPS-induced secretion of cytokines by macrophages is strongly suppressed by steroids [28].

#### 2.3.2. A549 Lung Carcinoma Cells

Human A549 lung carcinoma cells (ATCC^®^ CCL-185TM) were grown in DMEM supplemented with 10% FCS, 2 mM L-glutamine, penicillin at 100 U/mL, and streptomycin at 100 μg/mL and maintained at 37 °C in a humidified atmosphere of 95% air/5% CO_2_. The A549 cells were seeded at a density of 100,000 cells per well in 200 µL of DMEM with 10% FCS in 96 flat-bottomed tissue culture-grade well plates. At the following day, the medium was replaced with a fresh medium containing various concentrations of mometasone furoate, and the cells were incubated in the absence or presence of 2.5 ng/mL of human IL-1β (200-01B, Peprotech, Cranbury, NJ, USA) for 24 h. The level of the GM-CSF (Granulocyte-macrophage colony-stimulating factor) was measured by ELISA, as described below (Section 2.5). Kagoshima et al. [29] observed that mometasone furoate inhibited IL-1β-induced GM-CSF production in A549 cells, the same as Newton et al. [30].

#### 2.3.3. PLB-985 Myeloid Leukemia Cells

The PLB-985 myeloid leukemia cell line, kindly provided by Dr. Zvi Granot of The Hebrew University, Jerusalem, Israel, was grown in an RPMI (Roswell Park Memorial Institute) 1640 medium (R8758, Sigma, St. Louis, MO, USA) supplemented with 10% heat-inactivated FCS, 2 mM L-glutamine, 1 mM sodium pyruvate (03-042-1B, Biological Industries, Beit HaEmek, Israel), 100 U/mL penicillin, and 100 μg/mL streptomycin. The cells were then differentiated into neutrophil-like cells by incubating 400,000 cells/mL for 5–7 days in the RPMI-1640 supplemented with 2.5% FCS, 0.5% N,N-dimethylformamide (DMF, 227056, Sigma, St. Louis, MO, USA), 2 mM L-glutamine, 1 mM sodium pyruvate, 100 U/mL penicillin, and 100 μg/mL streptomycin at 37 °C and 5% CO_2_ [31].

The differentiated neutrophil-like cells were seeded in Hank’s balanced salt solution (HBSS; 02-016-1A, Biological Industries, Beit HaEmek, Israel) at a density of 1,000,000 cells per well in 96-well plates in the absence or presence of different concentrations of mometasone furoate. Human TNFα (300-01A, Peprotech, Cranbury, NJ, USA) was added to a final concentration of 10 ng/mL or 100 ng/mL, and immediately thereafter, the reactive oxygen species (ROS) production was measured each minute for 1 h by reading the luminescence in a Tecan M200 infinite plate reader (Tecan Trading AG, Männedorf, Switzerland) after adding luminol (A8511, Sigma, St. Louis, MO, USA) to a final concentration of 50 µM and horseradish peroxidase (P8375, Sigma, St. Louis, MO, USA) to a final concentration of 4 U/mL [32,33].

### 2.4. Measuring the Biological Activity of the MMF Released from the Coated Stent

To study the anti-inflammatory activity of MMF released from the varnish-coated stents, the SNS segments coated with either SRV-MMF or SRV-placebo were immersed in Eppendorf tubes containing 2 mL of DMEM, followed by a 24 h incubation period at 37 °C. The segments were transferred daily to new Eppendorf tubes with 2 mL of fresh DMEM for a total of 20 days. To study whether the amount of MMF released from the coated stents had sufficient anti-inflammatory activity, the macrophage cell line, RAW 264.7, that was seeded in 96-well tissue culture plates, as described in Section 2.3.1, was exposed to 100 µL of the collected supernatants for one hour prior to the addition of 100 µL of LPS in DMEM with 10% FCS to obtain a final concentration of 10 ng/mL LPS. After a 24 h incubation, the levels of TNFα, IL-10, and IL-6 were determined using the respective ELISA kits (Peprotech, Cranbury, NJ, USA), as described in Section 2.5. It is well-known that the LPS-induced secretion of cytokines by macrophages is strongly suppressed by steroids [29].

### 2.5. Enzyme-Linked Immunosorbent Assay (ELISA) of Cytokines

The TNFα, IL-10, IL-6, and GM-CSF levels in supernatants of macrophages or A549 cells were measured using the respective standard TMB/ABTS ELISA developmental kits of Peprotech (Cranbury, NJ, USA) according to the manufacturer’s instructions. The detection ranges were: 32–2000 pg/mL for mouse TNFα (Murine TNF-alpha Mini TMB ELISA Development Kit, 900-TM54), 20–2000 pg/mL for mouse IL-10 (Murine IL-10 Mini ABTS ELISA Development Kit, 900-M53), 8–1000 pg/mL for mouse IL-6 (Murine IL-6 Mini ABTS ELISA Development Kit, 900-M50), and 10–1000 pg/mL for human GM-CSF (Human GM-CSF Mini ABTS ELISA Development Kit, 900-M30). In brief, the capture antibody was diluted to recommended concentrations in PBS, and 100 µL of this solution was added to each well of an ELISA Maxisorb Nunc-immune plate (442404, ThermoScientific, Roskilde, Denmark) for overnight incubation at room temperature. The wells were then washed four times with phosphate-buffered saline (PBS) containing 0.05% Tween-20 (170-6531, Bio-Rad, Hercules, CA, USA), blocked with 300 µL PBS containing 1% bovine serum albumin (BSA Fraction V; 0332-TAM, VWR Chemicals, Solon, OH, USA) for 1 h, followed by another four washes with PBS containing 0.05% Tween-20. After removing all the fluid in the wells, 100 µL of the samples or standard curve samples were added to the wells, followed by a 2 h incubation at room temperature. This was followed by six washes with PBS containing 0.05% Tween-20 and a 2 h incubation with a recommended dilution of the detection antibody in PBS containing 0.05% Tween-20 and 0.1% BSA. The wells were again washed with PBS containing 0.05% Tween-20 and incubated with recommended concentrations of streptavidin-HRP (TMB kit) or an avidin-HRP conjugate (ABTS kit) for 30 min. At the end of incubation, the wells were washed with PBS containing 0.05% Tween-20, and 100 µL 3,3′,5,5′-tetramethylbenzidine (TMB; T8665, Sigma, St. Louis, MO, USA) or 100 µL of an ABTS liquid substrate (EBK-ABTS, Peprotech, Cranbury, NJ, USA) was added to each well. The absorbance of the blue color was measured at 450 nm for TMB substrate and at 405 nm for ABTS substrate using a Tecan M200 Infinite plate reader. The cytokine levels were calculated against a standard curve made from the respective cytokines.

### 2.6. Statistical Analysis

The data are presented herein as the average ± standard deviation from 3 independent experiments, using a 1 cm-coated stent in each experiment. Statistical analysis was performed using Microsoft Excel software, Version 2302. A Student’s *t*-test was used to compare MMF-coated stents with placebo-coated stents, with a *p*-value less than 0.05 considered significant.

## 3. Results

### 3.1. Anti-Inflammatory Properties of Mometasone Furoate (MMF)

Initially, the studied anti-inflammatory activities of MMF were studied using three different cellular systems in vitro in order to find the one that could be used to study the slow release of MMF from the SRV-MMF-coated SNS (Figure 1A,B) and to enable the determination of whether the amount of MMF released from this varnish is sufficient to exert the desired effects.

#### 3.1.1. The Anti-Inflammatory Activity of Mometasone Furoate on PLB-985 Neutrophil-like Cells

The human PLB-985 leukemia cell line can be differentiated in vitro into neutrophil-like cells, which produce reactive oxygen species (ROS) in response to TNFα [33]. We wanted to study whether MMF could interfere with ROS production. To this end, differentiated PLB-985 cells were exposed to increasing concentrations of MMF in the absence or presence of 10 ng/mL TNFα (Figure 2A) or 100 ng/mL TNFα (Figure 2B), and the ROS production was followed by luminescence after adding luminol together with horseradish peroxidase (HRP). Both concentrations of TNFα induced ROS production in the differentiated PLB-985 cells (Figure 2A,B). When using 10 ng/mL TNFα, MMF at 10 µg/mL slightly inhibited TNFα-induced ROS production (Figure 2A), while at 100 ng/mL TNFα, this concentration of MMF had no inhibitory effect (Figure 2B). At 0.1 µg/mL MMF, there was even an increase in ROS production in the presence of TNFα (Figure 2A,B). Since MMF did not exert a strong inhibition on TNFα-induced ROS production (Figure 2A,B), this system could not be used for the testing of the SRV-MMF-coated stents.

#### 3.1.2. The Inhibitory Effect of Mometasone Furoate on IL-1β-Induced GM-CSF Production in A549 Lung Carcinoma Cells

Another cell system that has been reported to respond to MMF is the A549 lung carcinoma cells [30]. Previous studies have documented that MMF could prevent the IL-1β-induced GM-CSF production in these cells [30], which is one axis in the inflammatory process [34]. We, therefore, tested if this system is sufficiently sensitive to MMF. To this end, confluent A549 cell layers were exposed to increasing concentrations of MMF in DMEM containing 1% FCS, followed by exposure to 2.5 ng/mL IL-1β. The following day, the GM-CSF levels were analyzed by an ELISA. MMF at the highest concentration tested (10 μg/mL) did not prevent the IL-1β-induced GM-CSF. However, at the lower concentrations of 0.1–1 μg/mL of MMF, there was a 60–70% inhibition (Figure 3). Although the response to MMF was significant, the GM-CSF levels were relatively low in the range of pg/mL, making this system not sufficiently sensitive for our aim.

#### 3.1.3. The Anti-Inflammatory Activity of Mometasone Furoate on RAW264.7 Macrophages

We next studied the effect of MMF on the LPS-induced secretion of TNFα, IL-6, and IL-10 by the RAW264.7 macrophage cell line. Steroids are well-known to inhibit macrophage function [35]. To this end, RAW264.7 macrophages were exposed to various concentrations of MMF and 10 ng/mL of lipopolysaccharide (LPS). After a 24 h incubation, the TNFα and IL-6 levels in the supernatants were analyzed by an ELISA. MMF was found to partly inhibit the LPS-induced TNFα secretion (Figure 4A), while it had a more profound inhibitory effect on IL-6 secretion (Figure 4B). These data suggest that IL-6 production by RAW264.7 cells could be a good parameter to follow the anti-inflammatory effect of SRV-MMF-coated SNS.

### 3.2. The Anti-Inflammatory Activity of Mometasone Furoate Released from SRV-MMF-Coated Sinonasal Stents

We have designed a varnish containing MMF together with biocompatible polymers that form a film with sustained-release properties upon the evaporation of ethanol. This technique has previously been used with other compounds [36]. The coating was performed on the closed stent, and the varnish was still firmly adhered to the stent upon its opening, as shown in Cataldo et al. [37], which is required for better positioning of the stent in the nasal cavity. Both the plastic and metallic parts were coated with the varnish. We were interested to know for what extended time frame a release of MMF from the coated nasal stents could be achieved and if the released MMF was biologically active over such a timeframe. The biological activity of the released MMF was tested on LPS-stimulated macrophages, and the secreted cytokine levels were measured using commercial ELISA development kits.

The SRV-MMF or SRV-placebo coated stents were daily transferred daily to 1 mL of a fresh DMEM in Eppendorf tubes for a period of 20 days. The RAW 264.7 macrophages were exposed to 100 µL of the collected supernatants, to which 100 µL of LPS was added to a final concentration of 10 ng/mL. The controls were untreated macrophages, macrophages treated with LPS alone, and macrophages treated with MMF and LPS. The IL-6, IL-10, and TNFα levels were analyzed by ELISAs.

Supernatants collected from the SRV-MMF-coated SNS showed significant inhibition of LPS-induced IL-6 secretion up to day 14 to levels that were even less than those achieved with 100 ng/mL of MMF (Figure 5A). Thereafter, there were variations in the IL-6 levels, but these were still lower than those caused by 10 ng/mL of LPS (Figure 5A), indicating that a residual inhibitory effect was observed up to day 20 (Figure 5A). There were some individual variations from days 14 to 20. Notably, the supernatants collected from the SRV-placebo-coated SNS also had an inhibitory effect on IL-6 secretion at days 2 and 3, with minor effects thereafter (Figure 5A), indicating that some of the components of the varnish might contribute to the inhibitory effect. Both the SRV-MMF and SRV-placebo inhibited TNFα secretion, with a stronger effect of the SRV-MMF than the SRV-placebo, reaching the level observed with 100 ng/mL of MMF up to day 14 (Figure 5B), suggesting that the anti-inflammatory effect is caused by a combination of MMF and the polymers of the varnish. Similarly, both the SRV-MMF and SRV-placebo inhibited IL-10 secretion, with a stronger effect of the SRV-MMF than the SRV-placebo, reaching the level observed with 100 ng/mL of MMF up to day 17 (Figure 5B). Figure 6 shows the respective IL-6, TNFα, and IL-10 levels secreted by the same macrophages used in Figure 5 in the absence or presence of LPS and/or MMF. MMF at 100 ng/mL strongly inhibited LPS-induced IL-6 (Figure 6A) and IL-10 (Figure 6C) secretion while having only a partial inhibitory effect on TNFα secretion (Figure 6B).

## 4. Discussion

The American Association of Otolaryngology—Head and Neck Surgery recommends topical therapy, such as nasal rinses, as the first-line treatment for CRS [38]. These rinses can be conducted alone or in combination with a topical steroid, resulting in improved symptom burden [39]. The main problem of topical therapy is low patient compliance, which has been shown to be less than 25% and 50% for saline irrigation and intranasal steroids, respectively [40].

SNS are frequently inserted into the nasal cavity following endoscopic sinus surgery (ESS) to prevent restenosis and scarring of the neo-ostium, helping to keep the sinus outflow tract. The first steroid-eluting sinus stent with mometasone furoate (MMF) was introduced in 2011 with FDA approval [41]. Since the early trials, MMF-eluding stents have shown positive outcomes for patients with CRSwNP over a range of subjective and objective endpoints, such as reduced nasal obstruction/congestion/olfactory loss scores, polyp shrinkage, and a reduction in the need for sinus surgery by 61% [8].

Herein, we proposed that coating these SNS with an SRV containing steroids such as MMF would be productive in reducing mucosa inflammation. It was important to study whether the coating could be retained on the stents for long periods of time with a concomitant daily release of a sufficient amount of steroids to the surroundings to reduce mucosal inflammation.

Different types of drugs can be incorporated into formulations, and different types of devices can be coated with this varnish. Several devices with SRV have already been developed, tested, and commercialized in different medical fields, especially in dentistry [42,43,44]. A pharmacological advantage of SRV is that it prolongs the time of release and subsequently extends the residence of the drug. Due to this, it requires a lower dose of the drug to produce a response, and thus potential side effects are reduced, improving its safety. In contrast, traditional nasal delivery devices (local drops, creams, and sprays) have a rapid release of the active agent and rapid elimination rate of the agent from the nasal mucosa, which is not time-controlled [42,44]. We previously reported some encouraging results with a similar sustained-release system where chlorhexidine was released at active anti-bacterial levels for up to 25 days [37].

In the present study, we wanted first to find an in vitro system that could be used for studying the anti-inflammatory activity of steroids such as MMF that can be used for studying the efficacy of SRV-MMF-coated SNS. Ideally, it would have been best to use primary eosinophils, which are important white blood cells involved in the inflammation of the sinonasal cavity [16]. However, the amount of primary eosinophils that can be isolated is relatively low, which is not sufficient for a comprehensive study. Therefore, we tried other systems, including differentiated PLB-985 neutrophil-like cells, A549 epithelial lung carcinoma cells, and RAW264.7 macrophages. We observed that MMF did not affect the TNFα-induced ROS production in the differentiated PLB-985 cells but inhibited both the IL-1β-induced GM-CSF secretion by the A549 cells and cytokine secretion (IL-6, TNFα, and IL-10) by the macrophages. MMF significantly inhibited IL-6 and IL-10 secretion by the macrophages, while its effect on TNFα secretion was only partial. In support of our findings, it has been reported that MMF inhibits LPS-induced IL-1, IL-6, and TNFα expressions in mouse blood cells [45].

In our in vitro model of the LPS-induced secretion of TNFα, IL-6, and IL-10 by the RAW264.7 macrophage cell line, SRV-MMF had an inhibitory effect on the secretion of IL-6, TNFα, and IL-10 for a time period of 14 to 17 days. The inhibitory effect of SRV-MMF-coated SNS suggests that this SRV drug system may decrease inflammation in the upper airways by inhibiting pro-inflammatory cytokine release and, consequently, lead to a reduction in inflammatory cell recruitment and activation promoted by such cytokines.

The reason that the SRV-placebo occasionally suppressed macrophage secretion of the tested cytokines could be due to the slow release of the polymers that constitute the varnish, thereby affecting macrophage function. Indeed, the low molecular weight PEG has been shown to reduce inflammatory cytokine expression in LPS and zymosan models of sepsis [46]. The immunosuppressive properties of PEG-400 did not interfere with our aim and, in fact, may have contributed to attenuating the activity of inflammatory macrophages.

In vitro biological release kinetics is an important aspect of the characterization of SRV delivery systems, especially when dealing with unique organs such as the nasal cavity. Biological release kinetic assessments of the active agent can lead to a better understanding of the system and of ways to improve and better control its pharmacological behavior [47].

Nasal drops, nasal sprays, and nasal irrigation have the advantages of being simple and economical but have present disadvantages, such as patient compliance, inaccurate dosage, and difficulty in reaching the depth of the nasal cavity. Biomaterials and sinus implants have provided more durable and effective nasal drug delivery post-surgery [48]. The use of such slow-release devices proposed herein has the advantage that the drug is delivered locally, with sustained, low concentrations, thereby extending the time of action and reducing the toxicity of single high dosages. Other antimicrobial agents or anti-inflammatory agents or a combination of them (e.g., steroids and CHX) may be used for the formulation of additional versions of novel SRVs.

Commercially available stents such as PROPEL and SINUVA are absorbable and already come with the drug incorporated into the stent, making these products expensive and even inaccessible for several countries. The advantage of our SRV technology is that the varnish comes separately for the stent and can be applied on different types of nasal stents that are less expensive. The flexibility of our SRV makes it possible to incorporate any desired combinations of drugs according to need.

In this study, we have proven the concept that SRV technology for CRS can be used in in vitro models. Based on these results, in vivo clinical trials are indicated to further evaluate the in vivo efficiency of this technique.

## 5. Conclusions

In this study, we have successfully demonstrated that SRV-MMF-coated SNS had an inhibitory effect on the secretion of IL-6, TNFα, and IL-10 for a time period of 14 to 17 days using the in vitro RAW264.7 macrophage model. The inhibitory effect of the SRV-MMF suggests that the SRV drug system can be used to decrease inflammation in the upper airways by inhibiting pro-inflammatory cytokine release, leading to a reduction in inflammatory cell recruitment and activation promoted by such cytokines. These promising results may play a significant role in the future treatment options available for chronic rhinosinusitis, and using this SRV-MMF technology applied to different nasal stents or other nasal drug delivery devices may reduce the need for revision endoscopic sinus surgery in patients with chronic sinusitis. Further studies evaluating the efficacy of the SNS’ SRV-MMF in pre-clinical models are needed.

## Figures and Tables

**Figure 1 pharmaceutics-15-01015-f001:**
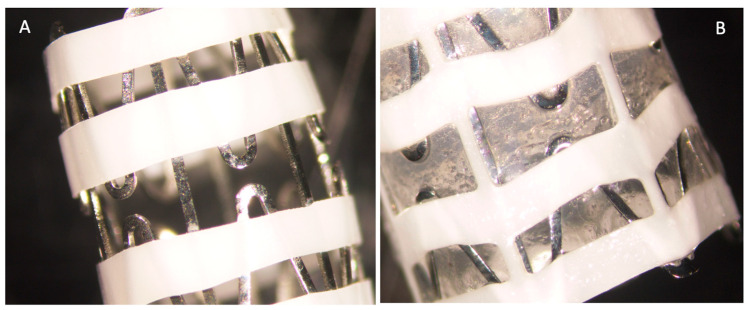
(**A**) SNS without varnish. (**B**) SRV-MMF-coated stent. The coating was retained on both the polyurethane plastic material and the nitinol wires.

**Figure 2 pharmaceutics-15-01015-f002:**
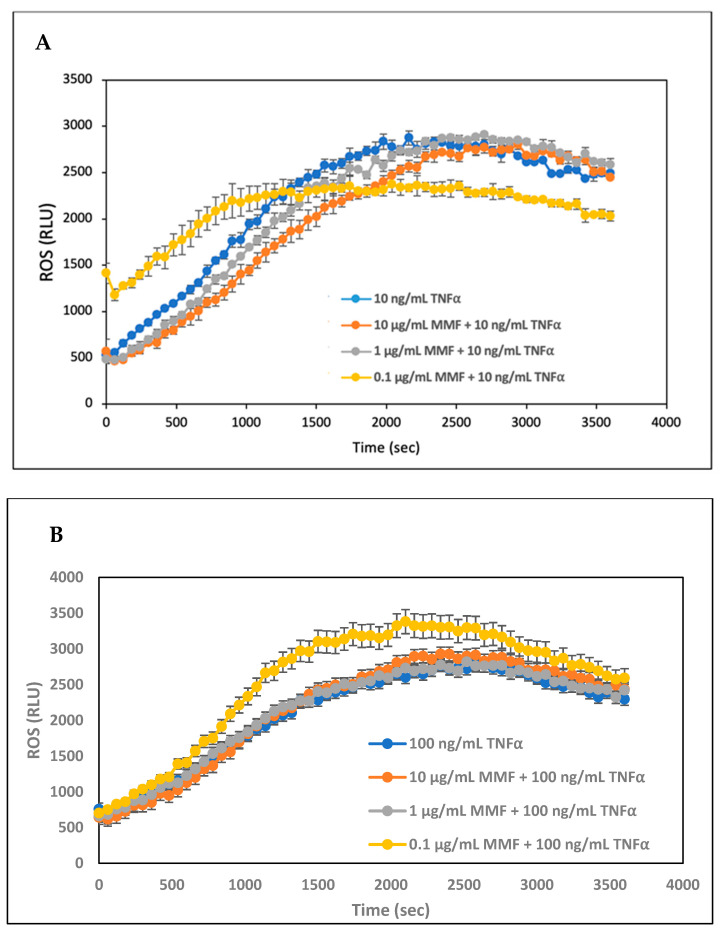
(**A**) DMF-differentiated PLB-985 cells were exposed to different concentrations of MMF (0.1, 1.0, and 10 µg/mL) and/or 10 ng/mL TNFα, and the ROS production was followed by monitoring the luminescence each minute for 60 min. MMF only slightly affected the TNFα-induced ROS production. (**B**) The same setup experiment as in (**A**) but with 100 ng/mL TNFα. At this TNFα concentration, the presence of 1.0 and 10 µg/mL MMF did not interfere with the TNFα-induced ROS production, and 0.1 µg/mL MMF even increased the TNFα-induced ROS production. RLU = Relative luminescence units. The values are the average of triplicates, and the standard deviation is presented.

**Figure 3 pharmaceutics-15-01015-f003:**
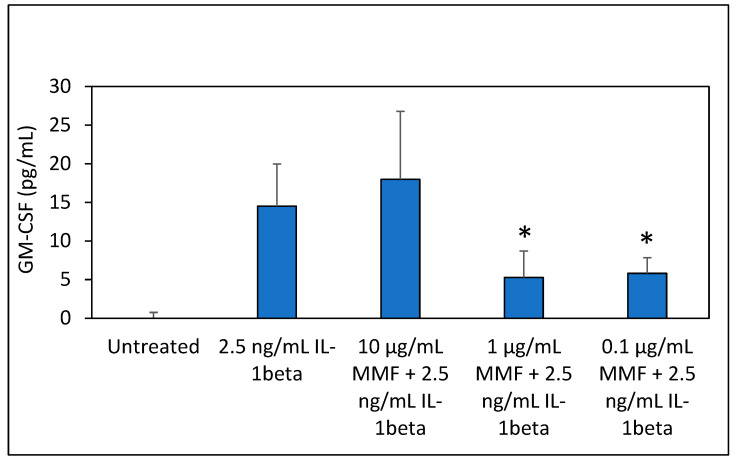
A549 lung carcinoma cells were exposed to different concentrations of MMF (0.1, 1 µg/mL and 10 µg/mL) and 2.5 ng/mL IL-1β for 24 h. The GM-CSF levels in the supernatant were determined by an ELISA. MMF suppressed the IL-1β-induced GM-CSF production in A549 lung cancer cells. The lowest concentrations of mometasone (0.1–1 µg/mL) caused strong and similar inhibition of GM-CSF production, while the higher concentration of 10 µg/mL MMF had no effect. The values are the average of triplicates, and the standard deviation is presented. * *p* < 0.05.

**Figure 4 pharmaceutics-15-01015-f004:**
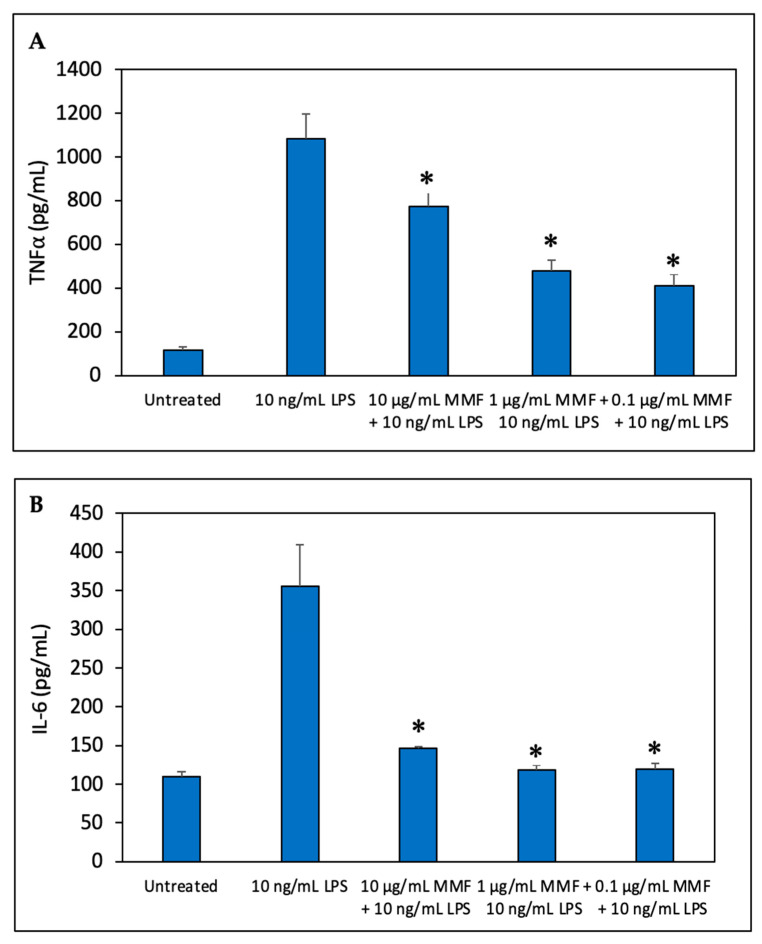
A cellular model of inflammation. RAW264.7 macrophages were exposed to different concentrations of MMF (0.1, 1 µg/mL and 10 µg/mL) in the presence of 10 ng/mL LPS for 24 h. The levels of TNFα (**A**) and IL-6 (**B**) in the macrophage supernatants were determined by an ELISA. Mometasone suppressed both TNFα and IL-6 secretion from LPS-exposed macrophages. The macrophages produced a low amount of TNFα and IL-6 in the absence of LPS, which was strongly induced in its presence. The lower tested concentrations of MMF (0.1–1 µg/mL) caused stronger inhibition of TNFα than the higher one (10 µg/mL). The values are the average of triplicates, and the standard deviation is presented. * *p* < 0.05 compared to LPS alone.

**Figure 5 pharmaceutics-15-01015-f005:**
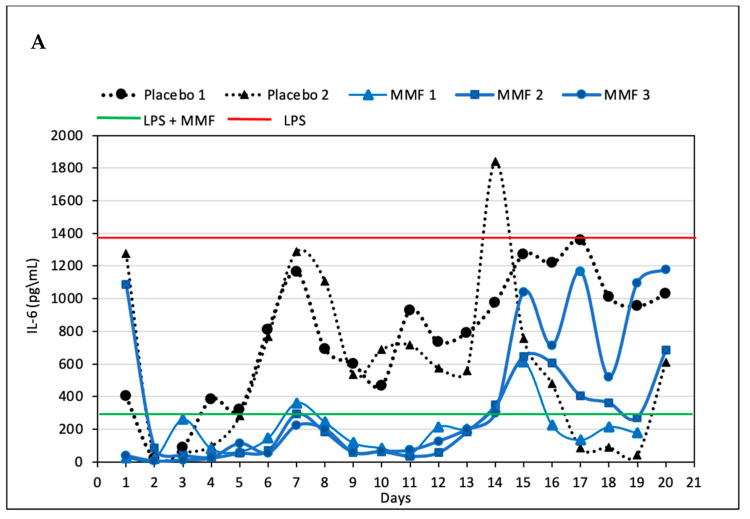
A cellular model of inflammation: SRV-MMF and SRV-placebo-coated nasal stents were incubated in 2 mL DMEM for 24 h and daily transferred to a fresh DMEM for 20 days. Then, the RAW 264.7 macrophages were incubated with the supernatant in the presence of LPS (10 ng/mL) for 24 h, and the secreted IL-6 (**A**), TNFα (**B**), and IL-10 (**C**) levels were analyzed by an ELISA. The red line presents the IL-6, IL-10, and TNFα levels secreted by the macrophages in the presence of 10 ng/mL LPS, while the green line presents their levels in the macrophage samples exposed to LPS and 100 ng/mL MMF. The black lines are the SRV-placebo-coated samples of two individual samples, while the blue lines are the SRV-MMF-coated samples of three individual samples. Figure 6A,C shows the IL-6, TNFα, and IL-10 levels, respectively, secreted from the macrophages in the same experiment in the absence or presence of 10 ng/mL LPS and/or MMF. The inhibitory effect of the SRV-MMF-coated stents on the secretion of all three cytokines was statistically significant (*p* < 0.05), while the SRV-placebo had only a significant inhibition on IL-10 secretion (*p* < 0.05).

**Figure 6 pharmaceutics-15-01015-f006:**
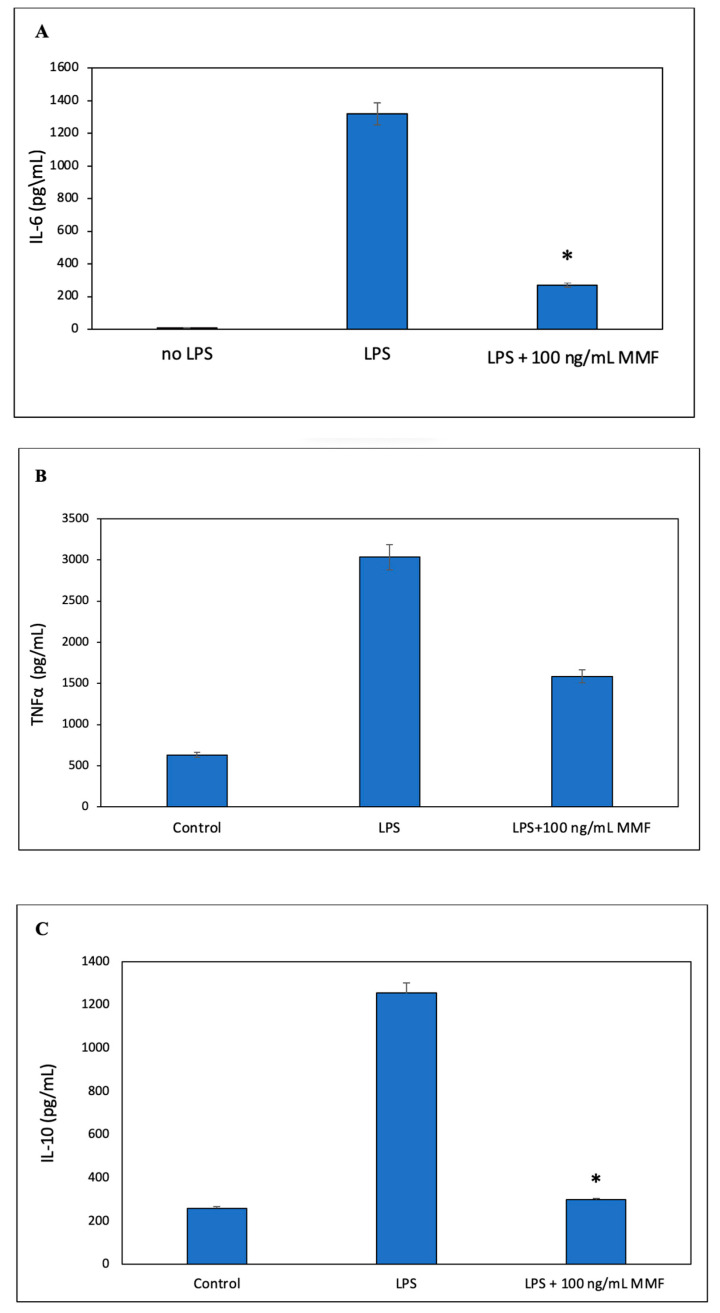
The control samples of data presented in Figure 5. The same macrophages that were used for measuring the anti-inflammatory action of supernatants that have been exposed to either the SRV-placebo or SRV-MMF-coated SNS were exposed to 10 ng/mL LPS alone or in the presence of 100 ng/mL MMF. The IL-6 (**A**), TNFα (**B**), and IL-10 (**C**) levels were measured simultaneously with the samples in Figure 5 by an ELISA. MMF significantly prevented LPS-induced IL-6 secretion (**A**) and IL-10 (**C**), with only limited inhibition of TNFα secretion (**B**). The values are the average of triplicates, and the standard deviation is presented. * *p* < 0.05. The IL-6, IL-10, and TNFα levels of untreated (“control”) macrophages are also shown.

## Data Availability

Availability of data and material: the datasets generated during and/or analyzed during the current study are available from the corresponding author on reasonable request.

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
