# Peer review of "Sinonasal Stent Coated with Sustained-Release Varnish of Mometasone Furoate Inhibits Pro-Inflammatory Cytokine Release from Macrophages: An In Vitro Study"

_pharmaceutics, 2023, doi:10.3390/pharmaceutics15031015_

Round 1

Reviewer 1 Report

The Authors of “Sinonasal Stent Coated with Sustained-Release Varnish of Mometasone Furoate Inhibited Pro-Inflammatory Cytokine Release from Macrophages: An in vitro Study” tested a varnish releasing mometasone furoate for the ability to reduce the inflammation. It is a nice study demonstrating positive and neutral outcomes of using MMF applied via SRV. My main concern is understanding the added value of this work since two FDA-approved systems using the same drug are already on the market. 

Study justification: What would be the expected advantage of using SRV over PROPEL or SINUVA? Both use the same active compound. Is it cost, the difficulty level of preparation, or perhaps something else? Please add to the Introduction and Discussion.

Methods

Please decide on consequently using one format when reporting the cell concentration (1 x 106 or 1,000,000).

Please provide the catalog numbers of the cell culture plastic, reagents, and equipment used for the experiments.

 “horse radish” should be spelled “horseradish.”

Results

Figure 1 – please specify if and if yes, how often this experiment was repeated and if the duplicates/triplicates were used. The same applies to Fig. 2 and 3. Do the values represent means or medians? Is it SD or SE that is plotted?

Figure 4 – please specify if and if yes, how many times this experiment was repeated and if the duplicates/triplicates were used. There are no error bars this time – does it mean that the experiment was performed only once with single samples?

Figure 5 – Based on the presented data, it can be assumed that the Sustained-Release Varnish itself has inhibitory properties regarding cytokine release. Are there any statistical data on that? I can see that being adequately addressed in the Discussion, and my question relates purely to data analyses. 

Reviewer 2 Report

pharmaceutics-2256830-peer-review-v1

The paper is interesting and clear with potential applications. In my opinion the project was presented well. Maybe negative point is the fact that authors have not applied as control already existed two commercial approaches. Why not?

In general, the paper is well structured and presented.

The introduction is well prepared and structured, presenting principal points on existing and suggesting new ways of approaching the problem, including stating benefits from selecting the MMF.

Do authors have any controls? Why commercial stents already mentioned in the introduction were not used as controls?

Please, try to avoid discussion in the results section and not need repeating results in the discussion section.

In my opinion, discussion can be upgraded a bit more, and some additional comparative examples provided.

Reviewer 3 Report

This study is an in vitro study of a newly developed stent with MMF varnish. Although the study is novel, it is not clearly presented and need much improvement for publication.

- The title should be in present sentence: “MMF inhibits ~ “(not inhibited).

- Please provide the full term of LPS at first apprearance in the manuscript.

- ln line 61,use the word PROPEL consistantly

- What is the difference of this SRV and previous stents (PROPEL and SINUVA)?

- The description in section 3.1. (line 247-255) should be presented elsewhere, not in the results section.

- In Discussion, the abreviations are again narrated.

- There is no figure for the stent used in this study or the coatings of the varnish.

- How about the in vivo or animal study ?

Round 2

Reviewer 3 Report

The authors have answered to all the query that I had raised.